# Snapshot of the Phylogenetic Relationships among Avian Poxviruses Circulating in Portugal between 2017 and 2023

**DOI:** 10.3390/vetsci10120693

**Published:** 2023-12-07

**Authors:** Daniela Santos, Teresa Fagulha, Margarida Dias Duarte, Ana Duarte, Fernanda Ramos, Sílvia Carla Barros, Tiago Luís, Ana Margarida Henriques

**Affiliations:** 1Laboratório de Virologia, Instituto Nacional de Investigação Agrária e Veterinária, 2780-157 Oeiras, Portugal; dl.santos@campus.fct.unl.pt (D.S.); teresa.fagulha@iniav.pt (T.F.); margarida.duarte@iniav.pt (M.D.D.); ana.duarte@iniav.pt (A.D.); fernanda.ramos@iniav.pt (F.R.); silvia.santosbarros@iniav.pt (S.C.B.); tiago.luis@iniav.pt (T.L.); 2Centre for Interdisciplinary Research in Animal Health (CIISA), Faculty of Veterinary Medicine, University of Lisbon, 1300-477 Lisbon, Portugal; 3Associate Laboratory for Animal and Veterinary Sciences (AL4AnimalS), Faculty of Veterinary Medicine, University of Lisbon, 1300-477 Lisbon, Portugal

**Keywords:** Avipoxvirus, molecular characterization, phylogenetic analysis, viral introduction, portugal

## Abstract

**Simple Summary:**

Avipoxvirus, the causative agent of avian pox disease, can infect more than 278 species of wild and domestic birds. It causes multiple negative consequences for both the economy and the ecosystem. These include reduced egg production, reduced mating success, growth retardation, and occasionally death. Several diagnostic methods are currently available, and prevention is mainly performed by implementing sanitary measures and vaccination. In order to evaluate and characterize the Avipoxviruses circulating in Portugal between 2017 and 2023, ten samples positive for this virus were analyzed. They were then compared with other isolates from other hosts, countries, and years of collection. As a result, it was possible to understand that certain variants of the virus continued to circulate over the years despite the introduction of new viruses.

**Abstract:**

Avipoxvirus (APV), a linear dsDNA virus belonging to the subfamily *Chordopoxvirinae* of the family *Poxviridae*, infects more than 278 species of domestic and wild birds. It is responsible for causing avian pox disease, characterized by its cutaneous and diphtheric forms. With a high transmission capacity, it can cause high economic losses and damage to the ecosystem. Several diagnostic methods are available, and bird vaccination can be an effective preventive measure. Ten APV-positive samples were analyzed to update the molecular characterization and phylogenetic analysis of viruses isolated in Portugal between 2017 and 2023. A P4b gene fragment was amplified using a PCR, and the nucleotide sequence of the amplicons was determined using Sanger sequencing. The sequences obtained were aligned using ClustalW, and a maximum likelihood phylogenetic tree was constructed. With this study, it was possible to verify that the analyzed sequences are distributed in subclades A1, A2, B1, and B3. Since some of them are quite similar to others from different countries and obtained in different years, it is possible to conclude that there have been several viral introductions in Portugal. Finally, it was possible to successfully update the data on Avipoxviruses in Portugal.

## 1. Introduction

The family *Poxviridae* is divided into subfamilies *Chordopoxvirinae* and *Entomopoxvirinae*, responsible for infecting vertebrates and insects, respectively [1]. The genus Avipoxvirus is included in the subfamily *Chordopoxvirinae*, along with seventeen other genera [2]. Regarding the different bird species that the virus infects, there are twelve species of Avipoxviruses (APVs) [3].

According to Manarolla et al. [4], the nomenclature given to APVs is based on the type of species present in each clade, with five clades already identified, A to E. Clade A is associated with fowlpox, clade B with canarypox, and clade C with psittacinepox viruses [5]. To date, seven subclades within clade A and four subclades within clade B have been described. Clade D includes a unique strain, QP-241, which was isolated from a quail in Italy [6]. Clade E contains sequences isolated from APVs from chickens in Brazil [7] and Mozambique [8] and also from a turkey in Hungary [9]. These studies of phylogenetic classification were conducted with fragments of the genes encoding the 4b core-like protein (P4b) and the DNA polymerase, both highly conserved among poxviruses [10].

Avipoxviruses are very large viruses [11] that can be either oval or brick-shaped [12]. The dsDNA genome is in a linear configuration [13] and has a very low GC content [14]. Of all the poxviruses, Avipoxviruses have the largest genome, approximately 288–300 kbp long [15], and encode more than 320 putative genes [14]. The replicative cycle, unlike other virus species, is characterized by transcription carried out in the cytoplasm [1].

APV infection can lead to the development of avian pox disease, which can present in its cutaneous or diphtheric forms [16]. These are caused by different routes of infection. The cutaneous form is caused by mechanical trauma [17] and is responsible for nodular lesions in featherless areas, being associated with very low mortality rates [18]. The diphtheric form, on the other hand, occurs after inhalation or ingestion of the virus [17] and is characterized by the formation of proliferative nodular lesions in the mucous membranes of the digestive and respiratory tracts [19]. As the nodular lesions can cause breathing and eating difficulties, the mortality rate is higher in this form of the disease [20]. The severity of the clinical symptoms depends on the initial virulence and pathogenicity of the virus strain [21]. Nevertheless, the avian species, age [22], and the presence of secondary viral or bacterial infections are also important factors [21]. Transmission can occur via mechanical vectors such as arthropods, aerosols released by sick birds, direct contact with lesions, or ingestion of contaminated food or drink [22]. Wild bird behavior, such as migration, introduction of new species, and habitat change, can also lead to transmission [23]. However, the disease can be prevented by implementing sanitary measures to avoid mechanical vectors and contaminated sources [21] or vaccinating birds [24].

Infection leads to a number of negative consequences, both economic and ecological. A decrease in egg production and immunity is common, resulting in a reduced ability to survive secondary infections. Reduced mating success, growth retardation, blindness, feeding difficulties, and death may also occur [15,23,25]. As a result, poultry farmers suffer significant economic losses due to the need to replace livestock, lost sales, and sanitation costs [21,26]. At the ecosystem level, the birds are more vulnerable, leading to increased predation and reduced mating, resulting in population decline [27].

In this study, the molecular characterization and phylogenetic analysis of Avipoxviruses isolated from ten samples detected in Portugal between 2017 and 2023 are performed in order to infer the possible origin of the viruses circulating in the country.

## 2. Materials and Methods

### 2.1. Samples

Several samples arrive each year at INIAV for screening for Avipoxvirus infection. This screening is performed using the method referred to by Huw Lee and Hwa Lee [28] and described in Section 2.3. Between 2017 and 2023, ten positive samples were obtained (Table 1) and preserved at −80 °C. The positive results obtained were only communicated to the respective customers and were not published anywhere.

### 2.2. Nucleic Acids Extraction

The viral DNA extraction was performed in a nucleic acid extraction workstation, Kingfisher Flex (Thermo Fisher Scientific, Waltham, MA, USA), using the IndiMag Kit (Indical Bioscience, Leipzig, Germany), following the manufacturer’s instructions. After extraction, the DNA was stored at 4 °C.

### 2.3. PCR Amplification of the P4b Gene Fragment

The amplification of the P4b gene fragment was performed using conventional PCR with primers described by Huw Lee and Hwa Lee [28], using NZYTaq II 2× Green Master Mix (0.2 U/μL) (NZYTech, Lisboa, Portugal). The reaction contained 0.5 μL of each primer (50 pmol/μL) and 5 μL of DNA, for a total of 25 μL. The PCR program was executed in a UNO II thermocycler thermoblock (Biometra, Göttingen, Germany) and consisted of an initial denaturation at 95 °C for 2 min, followed by 50 cycles of denaturation at 95 °C for 30 s, annealing at 60 °C for 40 s, and extension at 72 °C for 1 min. A final extension was performed at 72 °C for 10 min. The PCR product was separated with a 1% agarose gel electrophoresis stained with GreenSafe Premium (NZYTech, Lisboa, Portugal). The expected fragments of 578 bp, corresponding to the P4b gene amplified fragment, were excised and purified using the NZYGelpure Kit (NZYTech, Lisboa, Portugal), according to the manufacturer’s instructions.

### 2.4. Sanger Sequencing of the P4b Gene Fragment

The P4b gene fragments were sequenced using the Sanger method, using the BigDye Terminator v3.1 Cycle Sequencing Kit (Thermo Fisher Scientific, MA, USA), according to the manufacturer’s guidelines. This fragment was chosen since it was mostly used in the phylogenetic analysis of these viruses, and the majority of the nucleotide sequences available in the database belong to this fragment. The primers used in the sequencing reaction are listed in Table 2. To improve the sequencing quality, different sample volumes were used in each reaction depending on the DNA concentration. Samples were sequenced in a 3130 Genetic Analyzer (Applied Biosystems, Waltham, MA, USA). Sequence alignment was performed with SeqScape Software from Applied Biosystems. Using Clustal Omega by EMBL-EBI (https://www.ebi.ac.uk/Tools/msa/clustalo, accessed on 9 March 2023), percent identity matrices were also generated between the sequences of subclades A1, A2, B1, and B3 and the sequences isolated and sequenced in this work. The GenBank accession numbers of the sequences obtained in this study are shown in Table 1.

### 2.5. Phylogenetic Analysis

To study the phylogenetic origin of the Avipoxviruses detected and their phylogenetic relationship with the sequences published in GenBank (Table 2), a multiple alignment was performed between them using the ClustalW (https://www.genome.jp/tools-bin/clustalw, accessed on 10 March 2023) method. Sequences from different countries, years, and clades were chosen in order to obtain a reliable analysis. The sequences used for phylogenetic analysis were shortened in order to have the same length as those from the literature.

A phylogenetic tree was generated using the maximum likelihood method, using MEGA 11 with the Tamura 3-parameter G + I [29], determined as the best fit model also by MEGA 11, and 1000 bootstrap replicates [30]. Neighbor-joining and Bayesian methods were also used to generate the phylogenetic trees.

## 3. Results and Discussion

### 3.1. PCR Amplification of the P4b Gene Fragment

Amplification of the 578 bp P4b gene fragment was confirmed via 1% agarose gel electrophoresis. The results obtained are shown in Appendix A Appendix A. The presence of the expected fragment can be observed in all lanes except lanes 3, 4, and 5, corresponding to samples 11612-19 (puffin), 37026-19 (canary), and 03779-20 (canary), respectively. Amplification of DNA from samples 37026-19 and 03779-20 obtained in a previous extraction procedure was achieved (lanes 11 and 12) (Appendix A Appendix A). A new PCR reaction resulted in the amplification of the sample 11612-19 (Appendix A Appendix A). The P4b gene fragment from samples P-09292-22 and 00917-23 was previously obtained.

### 3.2. Sequencing of the P4b Gene Fragment

Alignment of the forward and reverse nucleotide sequences yielded a nucleotide sequence of 538 bp for all samples, except for sample 03779-20 (canary), after removal of the primer sequences. Due to incomplete sequencing of the 3′ end, only 513 bp were obtained for this sample. This result can be explained by DNA damage, incorrect amplification, or insufficient purification.

The alignment of the sequences studied is presented in Figure 1. Figure 1A presents the nucleotide alignment, while the alignment of the deduced amino acid sequences is presented in Figure 1B. The analysis of the alignments is carried out in the following section, together with the phylogenetic analysis.

### 3.3. Phylogenetic Analysis

The Avipoxvirus nucleotide sequences obtained were phylogenetically analyzed using different algorithms, namely maximum likelihood, neighbor-joining, and Bayesian analysis. The three algorithms produced phylogenetic trees with similar topologies. However, the maximum likelihood tree gave a better resolution of the sequences and was therefore chosen (Figure 2). The phylogenetic tree obtained, including representatives of all subclades, shows that clades C and D share a common most recent ancestor, as observed in previous studies [21,22].

The phylogenetic analysis shows that the samples from Portugal are well distributed, with five samples belonging to subclade A1, one to subclade A2, three to subclade B1, and one to subclade B3. Moreover, their distribution is quite consistent with the nomenclature given since samples from *Gallus gallus* belong to the fowlpox clade (clade A), whereas samples from *Turdus merula* and *Serinus canaria*, both Passeriformes, belong to the canarypox clade (clade B). *Phoenicopterus ruber*, *Fratercula*, and *Spheniscidae* are neither fowl nor Passeriforme’s representatives. The viruses detected in the first two hosts belong to the fowlpox clade, whereas the penguinpox virus belongs to the canarypox clade.

Comparing the alignment (Figure 1) and the percentage identity matrix of the sequences obtained in this study (Table 3), several conclusions can be drawn. Sample 16735-20 (blackbird) is the most distinct, as it has several different nucleotides along the entire sequence. Therefore, it has a very low percentage of identity compared to all the other sequences, with the highest percentage of identity being only 81.60%. The nucleotide sequence of sample 24569-17 (Flamingo) is the second most distinct sequence, with a maximum identity percentage of 90.52%. It also contains an additional valine codon at position 23.

By comparing the alignment of all the sequences present in the phylogenetic tree, it is observed that sample 16735-20 (Blackbird) has two unique mutations. It contains a cytosine instead of a thymine in position nt 63 and a thymine instead of a guanine or an adenine at position nt 148. These mutations originate amino acid changes, namely for a threonine (T) and a serine (S), as referred to below.

Interestingly, when analyzing the percentage identity matrix between the sequences of the A1 subclade, it can be observed that the puffin (*Fratercula*) isolate sequence is not completely identical to any other sequence, but the percentages of identity are quite high, showing that the sequences are very similar.

On the other hand, all the chicken (*Gallus gallus*) isolates have an identity percentage of 100%, except for sample sequence 04482-20. However, the identity percentage is still very high, 99.63%, with only two nucleotides differing. It can also be observed that the sequences of the samples isolated from turkeys (*Meleagris*, Germany, 2001) and wild turkeys (*Meleagris gallopavo*, Iran, 2015) are also similar to those isolated from chickens. By analyzing the position of the remaining sequences in the phylogenetic tree, it is possible to verify the presence of samples isolated from chickens in subclades C and E. When comparing those of subclade A1 with these, it is found that the percentage of identity is quite low, around 74%, confirming the classification in different clades.

Regarding the sequences of subclade A2, it can be observed that the sequence of the sample isolated from the flamingo (*Phoenicopterus ruber*) is different from all the other sequences. However, the percentages are quite high, varying between 92.15% (pigeon) (*Columbia livia*, USA, 1995) and 99.63% (pigeon) (*Columbidae*, Tanzania, 2013). Furthermore, it can be seen that the two APVs isolated from flamingos in Portugal belong to different clades: the one from 2010 belongs to B2 and the one from 2017 to A2. In fact, the percentage of identity between them is only 75.84%. When they are aligned, it can be seen that the 2010 sequence has an extra isoleucine codon at position nt 24, while the 2017 sequence has an extra phenylalanine codon at position nt 60.

The analysis of the sequences of subclade B1 shows that the two canary (*Serinus canaria*) isolates from Portugal have the same nucleotide sequence. The canary’s isolate from Austria is not 100% identical to these canary sequences, but it is very similar, with a 99.81% identity percentage and only one nucleotide difference. The sequence of the penguin (*Spheniscidae*) isolate is 100% identical to the sequences from three other isolates: the golden eagle isolate (*Aquila chrysaetos*, Spain, 2000), the stone-curlew isolate (*Burhinidae*, United Arab Emirates, 1998), and the crossbill isolate (*Loxia curvirostra*, Spain, 1930). A comparison of the sequence of the Avipoxvirus isolated from the penguin in Portugal with that from Argentina shows that they belong to different subclades, B1 and A3, respectively. These two sequences show several mutations between them, with a percentage of identity of only 72.68%.

Comparing the sequences of subclade B3, it can be seen that the sequence isolated from the blackbird (*Turdus merula*) is not completely identical to any other sequence. However, the percentage of identity is quite high among all sequences (97.03% to 97.58%).

Concerning the deduced amino acid sequences, it is possible to distinguish two major groups, one including sequences from clade A (24569-17, 23045-18, 11612-19, 04482-20, P-08508-21, and 00917-23), and another including sequences from clade B (37026-19, 03779-20, 16735-20, and P-09292-22), as expected. The most divergent sequence is that from strain 16735-20 from subclade B3, as occurred already with the nucleotide sequences. Amino acid sequences from strains 37026-19, 03779-20, and P-09292-22, all from the subclade B1, are almost identical. Regarding sequences from clade A, the same is observed, with that from subclade A2 showing more amino acid differences than those from A1, which are almost similar. When comparing the amino acid alignment of all the sequences present in the phylogenetic tree, it can be seen that sample 16735-20 (Blackbird) also has two unique mutations: a threonine (T) at site 21 and a serine (S) at site 50, as a result of the unique nucleotide mutations already referred.

The results of this study indicate that the APVs circulating in Portugal in chickens and canaries are quite similar since the nucleotide sequences of the isolates have not undergone significant changes. However, the isolates from flamingos show the opposite situation, as the virus circulating in 2010 is very different from the one circulating in 2017. This suggests the presence of a second virus introduction in Portugal. Finally, since the nucleotide sequence of the isolate from penguins is very similar to that of the isolates from golden eagles, stone curlews, and crossbills, it is possible to hypothesize that there has been a third viral introduction in Portugal.

## 4. Conclusions

This study has updated the molecular and taxonomic data of Avipoxviruses in Portugal based on ten viruses obtained between 2017 and 2023.

In conclusion, it was possible to verify that the samples studied were widely distributed in the phylogenetic tree, with representatives in subclades A1, A2, B1, and B3. It was also observed that some isolates, such as those from chickens and canaries, suggest that the APV circulating in Portugal originates from the same virus. Other isolates, such as those from flamingo and penguin isolates, suggest new virus introductions since they are similar to other strains from different years and geographical origins. These data suggest that the viruses circulating in Portugal have distinct origins. Interestingly, the sequence from the blackbird isolate has two unique mutations compared with all the sequences used in the phylogenetic analysis.

The major drawback of this study is the small sampling size used. However, there was no access to other sequences since only ten positive samples arrived at the laboratory between 2017 and 2023. It would be very interesting to analyze positive sequences that will arrive in the future to go deeper into the molecular characterization and phylogenetic analysis of Avipoxviruses circulating in Portugal.

## Figures and Tables

**Figure 1 vetsci-10-00693-f001:**
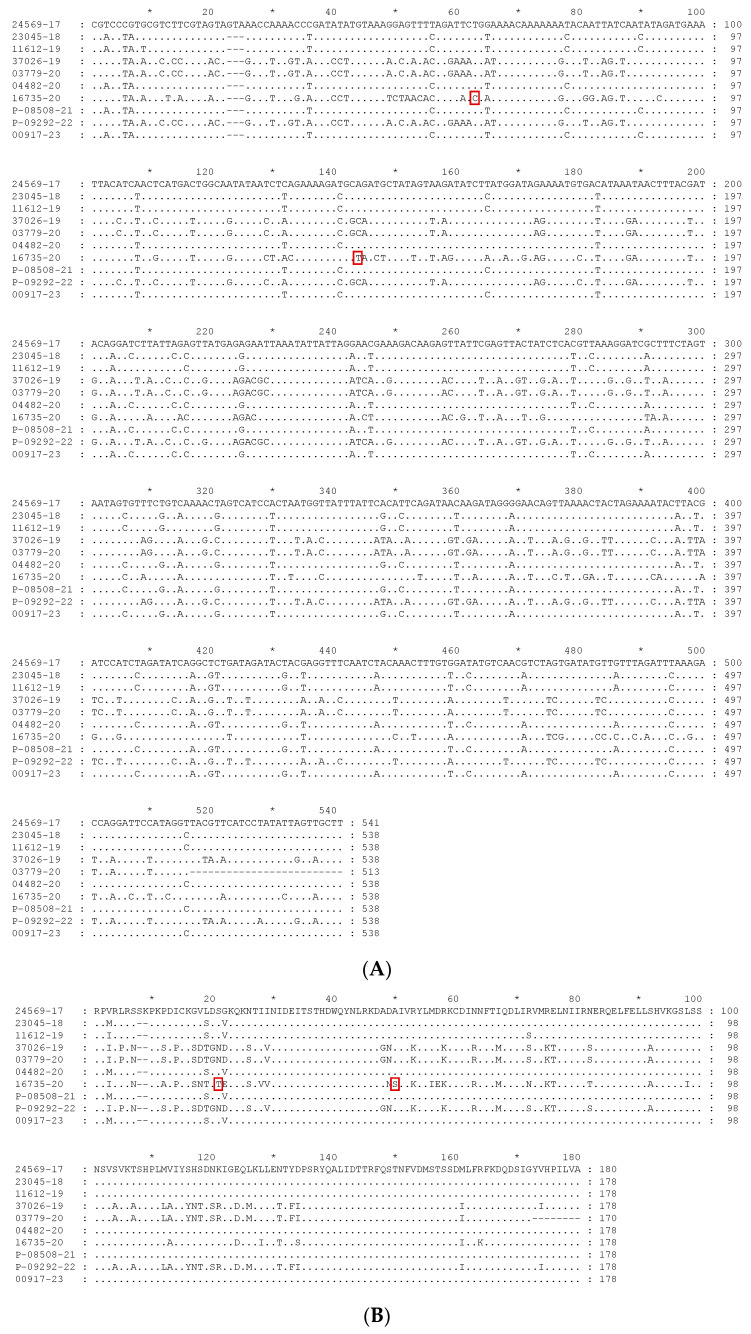
Alignment of the sequences obtained in this study. (**A**) Nucleotide sequences alignment; (**B**) deduced amino acid sequences alignment. Unique mutations observed in sequence 16735-20, when compared with all the sequences present in the phylogenetic tree, are heightened by a red square in both alignments.

**Figure 2 vetsci-10-00693-f002:**
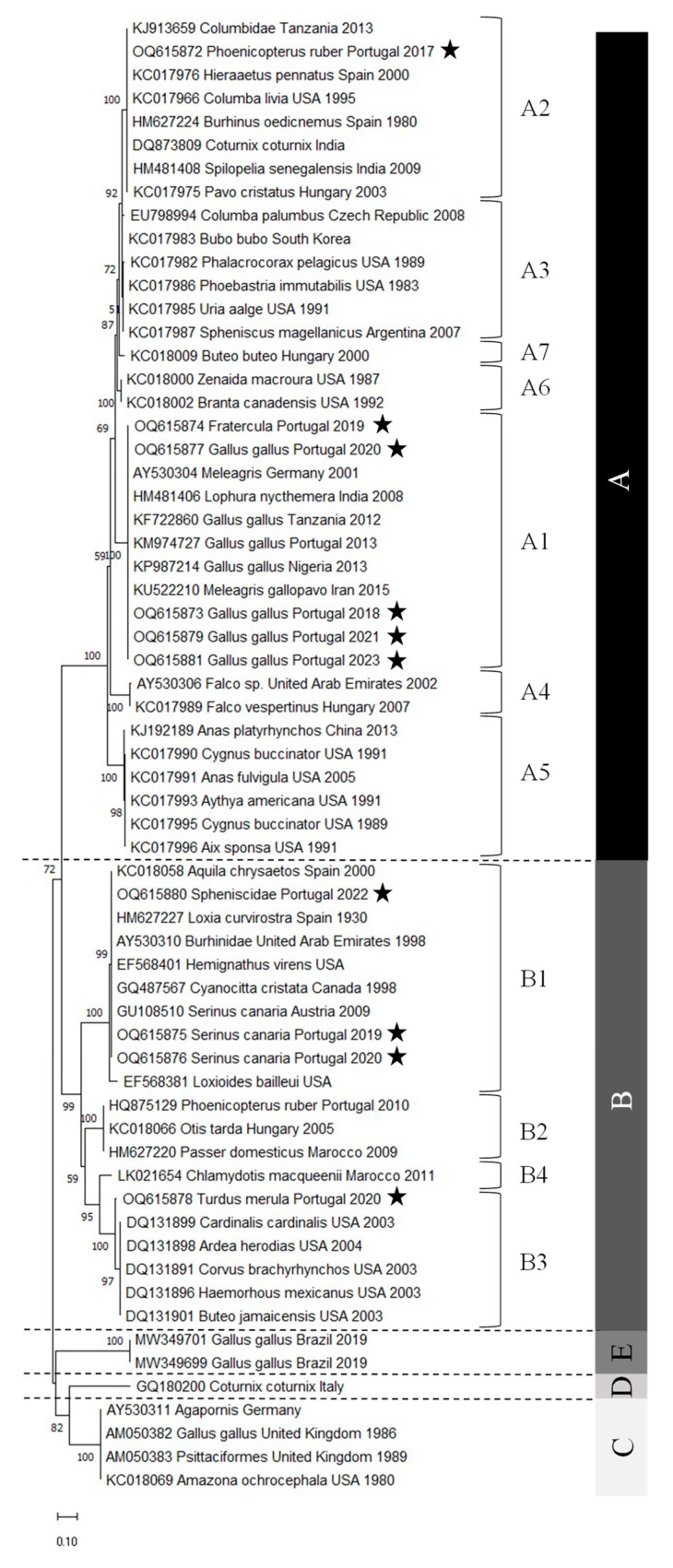
The evolutionary history was inferred using the maximum likelihood method and the Tamura 3-parameter model [29]. The tree with the highest log likelihood (−3244.07) is shown. The percentage of trees in which the associated taxa clustered together is shown next to the branches. Initial tree(s) for the heuristic search were obtained automatically by applying Neighbor-Join and BioNJ algorithms to a matrix of pairwise distances estimated using the Tamura 3 parameter model and then selecting the topology with superior log likelihood value. A discrete Gamma distribution was used to model evolutionary rate differences among sites (five categories (+G, parameter = 0.8553)). The rate variation model allowed some sites to be evolutionarily invariable ([+I], 23.87% sites). The tree is drawn to scale, with branch lengths measured in the number of substitutions per site. This analysis involved 63 nucleotide sequences. There was a total of 463 positions in the final dataset. Evolutionary analyses were conducted in MEGA11 [30]. Star pentagrams indicate the strains of this study. A, B, C, D and E characters indicate groups of strains belonging to clades A, B, C, D and E, respectively.

**Table 1 vetsci-10-00693-t001:** Avipoxvirus-positive samples were diagnosed at INIAV since 2017 and used in this study.

Host		Sequence ID	Accession Number	Country	Collection date	Isolation Source
Common Name	Scientific Name					
Flamingo	*Phoenicopterus ruber*	24569-17	OQ615872	Portugal: Lisboa	26 September 2017	Pool of organs
Chicken	*Gallus gallus*	23049-18	OQ615873	Portugal: Porto Santo, Madeira	24 July 2018	Pool of organs
Puffin	*Fratercula*	11612-19	OQ615874	Portugal: Lisboa	16 April 2019	Cutaneous lesion
Canary	*Serinus canaria*	37026-19	OQ615875	Portugal: Freixianda, Ourém	22 November 2019	Pool of organs
Canary	*Serinus canaria*	03779-20	OQ615876	Portugal	4 February 2020	Pool of organs
Chicken	*Gallus gallus*	04482-20	OQ615877	Portugal	11 February 2020	Cutaneous lesion
Blackbird	*Turdus merula*	16735-20	OQ615878	Portugal	9 June 2020	Cutaneous lesion
Chicken	*Gallus gallus*	P-08508-21	OQ615879	Portugal: Maia, Porto	22 September 2021	Pool of organs
Penguin	*Spheniscidae*	P-09292-22	OQ615880	Portugal: Avintes, Porto	17 October 2022	Pool of organs
Chicken	*Gallus gallus*	00917-23	OQ615881	Portugal: Évora	16 January 2023	Pool of organs

**Table 2 vetsci-10-00693-t002:** Details of APV sequences published in GenBank and used in this study.

Host		Accession Number	Country	Collection Date	Clade
Common Name	Scientific Name				
Turkey	*Meleagris gallopavo*	AY530304	Germany	2001	A1
Silver pheasant	*Lophura nycthemera*	HM481406	India	2008	A1
Chicken	*Gallus gallus*	KF722860	Tanzania	2012	A1
Chicken	*Gallus gallus*	KM974727	Portugal	2013	A1
Chicken	*Gallus gallus*	KP987214	Nigeria	2013	A1
Backyard turkey	*Meleagris gallopavo*	KU522210	Iran	2015	A1
Quail	*Coturnix coturnix*	DQ873809	India	-	A2
Indian little brown dove	*Spilopelia senegalensis*	HM481408	India	2009	A2
Eurasian stone-curlew	*Burhinus oedicnemus*	HM627224	Spain	1980	A2
Rock dove	*Columba livia*	KC017966	USA	1995	A2
Indian peafowl	*Pavo cristatus*	KC017975	Hungary	2003	A2
Booted eagle	*Hieraaetus pennatus*	KC017976	Spain	2000	A2
Pigeon	*Columbidae*	KJ913659	Tanzania	2013	A2
Wood-pigeon	*Columba palumbus*	EU798994	Czech Republic	2008	A3
Pelagic cormorant	*Phalacrocorax pelagicus*	KC017982	USA	1989	A3
Eurasian eagle owl	*Bubo bubo*	KC017983	Republic of Korea	-	A3
Common murre	*Uria aalge*	KC017985	USA	1991	A3
Laysan albatross	*Phoebastria immutabilis*	KC017986	USA	1983	A3
Magellanic penguin	*Spheniscus magellanicus*	KC017987	Argentina	2007	A3
Falcon	*Falco sp.*	AY530306	United Arab Emirates	2002	A4
Red-footed falcon	*Falco vespertinus*	KC017989	Hungary	2007	A4
Trumpeter swan	*Cygnus buccinator*	KC017990	USA	1991	A5
Mottled duck	*Anas fulvigula*	KC017991	USA	2005	A5
Redhead duck	*Aythya americana*	KC017993	USA	1991	A5
Trumpeter swan	*Cygnus buccinator*	KC017995	USA	1989	A5
Wood duck	*Aix sponsa*	KC017996	USA	1991	A5
Domestic mallard duck	*Anas platyrhynchos*	KJ192189	China	2013	A5
Mourning dove	*Zenaida macroura*	KC018000	USA	1987	A6
Canada goose	*Branta canadensis*	KC018002	USA	1992	A6
Common buzzard	*Buteo buteo*	KC018009	Hungary	2000	A7
Stone curlew	*Burhinidae*	AY530310	United Arab Emirates	1998	B1
Palila	*Loxioides bailleui*	EF568381	USA	-	B1
Amakihi	*Hemignathus virens*	EF568401	USA	-	B1
Blue jay	*Cyanocitta cristata*	GQ487567	Canada	1998	B1
Canary	*Serinus canaria*	GU108510	Austria	2009	B1
Red crossbill	*Loxia curvirostra*	HM627227	Spain	1930	B1
Golden eagle	*Aquila chrysaetos*	KC018058	Spain	2000	B1
House sparrow	*Passer domesticus*	HM627220	Morocco	2009	B2
Flamingo	*Phoenicopterus ruber*	HQ875129	Portugal	2010	B2
Great bustard	*Otis tarda*	KC018066	Hungary	2005	B2
American crow	*Corvus brachyrhynchos*	DQ131891	USA	2003	B3
House finch	*Haemorhous mexicanus*	DQ131896	USA	2003	B3
Great blue heron	*Ardea herodias*	DQ131898	USA	2004	B3
Northern cardinal	*Cardinalis cardinalis*	DQ131899	USA	2003	B3
Red-tailed hawk	*Buteo jamaicensis*	DQ131901	USA	2003	B3
MacQueen’s bustard	*Chlamydotis macqueenii*	LK021654	Marocco	2011	B4
Chicken	*Gallus gallus*	AM050382	United Kingdom	1986	C
Parrot	*Psittaciformes*	AM050383	United Kingdom	1989	C
Lovebird	*Agapornis*	AY530311	Germany	-	C
Yellow-crowned amazon	*Amazona ochrocephala*	KC018069	USA	1980	C
Quail	*Coturnix coturnix*	GQ180200	Italy	-	D
Chicken	*Gallus gallus*	MW349699	Brazil	2019	E
Chicken	*Gallus gallus*	MW349701	Brazil	2019	E
Domestic mallard duck	*Anas platyrhynchos*	KJ192189	China	2013	A5
Mourning dove	*Zenaida macroura*	KC018000	USA	1987	A6

**Table 3 vetsci-10-00693-t003:** Percent identity matrix between APV sequences isolated in this study.

		1	2	3	4	5	6	7	8	9	10
1	*Phoenicopterus ruber* (24569-17)		90.52	89.78	89.78	89.78	90.15	75.46	72.68	73.05	72.71
2	*Fratercula* (11612-19)			99.26	99.26	99.26	98.88	76.39	74.54	74.91	74.66
3	*Gallus gallus* (23049-18)				100.00	100.00	99.63	76.21	74.35	74.72	74.46
4	*Gallus gallus* (P-08508-21)					100.00	99.63	76.21	74.35	74.72	74.46
5	*Gallus gallus* (00917-23)						99.63	76.21	74.35	74.72	74.46
6	*Gallus gallus* (04482-20)							76.39	74.72	75.09	74.85
7	*Turdus merula* (16735-20)								81.60	81.60	81.48
8	*Spheniscidae* (P-09292-22)									99.63	99.81
9	*Serinus canaria* (37026-19)										100.00
10	*Serinus canaria* (03779-20)										

## Data Availability

Data are available upon request.

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
