# Peer review of "Snapshot of the Phylogenetic Relationships among Avian Poxviruses Circulating in Portugal between 2017 and 2023"

_vetsci, 2023, doi:10.3390/vetsci10120693_

Round 1

Reviewer 1 Report

Comments and Suggestions for Authors

The authors undertook to characterize phylogenetically avian poxviruses circulating in Portugal between 2017 and 2023. This is a valuable task but the publication unfortunately has a lot of drawbacks because much of the data is inaccurate or missing.

Specific comments:

Why P4b gene were chosen?, Please give additional information in Introduction related to genome, genes used in phylogenetic study of avian poxviruses? Whether this a single gene is sufficient for this type of molecular analysis?

Line 81 – ‘Those positive samples’

Please expand whether these are the tissue fragments described from Table 1, please cite Table 1.

Please specify by what diagnostic method these animals were found to be positive? Were the results regarding the diagnostics already described somewhere?

On what basis were the samples collected/obtained-no information on ethical issue.... in the paper

Line 92 - NZYTaq II 2x Green Master Mix – is this polymerase has proofreading activity? How many repeats of the PCR product were sequenced?

Line 119-121 – please check is this correct information? The best fit model is rather applied for particular methods NJ or ML or Bayesian….

Figure 1.- Any comments for double bands in lane 11 and 12? Maybe this is where the reason for the incomplete sequencing response lies – sample 03779-20 (Line 147)

What with samples P-09292-22 and 00917-23?

What the authors mean by “in a previous extraction” (Line 141)

Line 146-150 -If one of the sequences was 513bp long then the other theses analyzed should be shortened from 538bp in the phylogenetic analysis.

Figure 2. - The figure is presented in an unreadable way. As a rule, in such alignments, only dissimilar nucleotides are represented by 'letters' and identical ones by dots.

Line 161 – why B4 representatives are not present ?

Line 179-182 as previously the figure 2 is presented in an unreadable way. It will be recommended to highlight these unique mutations.

Line 200-203 – where is this alignment which showed these differences? The same comment to Line 218….

Line 245 – Figure 3 description contains ‘total of 485 positions’ when analyzed fragment was presented as 513/538bp; furthermore MEGA X is mentioned when Materials and Methods section contained MEGA 11 version (references MEGA X). This kind of percent identity matrix derived from multiple alignment of the nucleotide or/and the deduced amino acid sequences ( down/upper part of matrix). It seems that in both here is nucleotide one, so should not be repeated.

Comment to Results and Discussion section: Generally in this kind of work besides nucleotide alignment also deduced amino acids were also aligned. Why Authors did not present and discuss that???

Results and Discussion section contain only two citation – 5 and 14 – Line 162 ???

Missing information as: Author contributions, Funding, Institutional Review Board Statement.

Reviewer 2 Report

Comments and Suggestions for Authors

Introduction

While this manuscript is quite technical and specific, I believe it would be beneficial to have a broader introduction than what it currently included in the manuscript. I believe more information on APV infection, discussion on symptoms and consequences, would be beneficial. At the moment, it seems that they are simply named and accompanied by the appropriate citation, without including too much content. 

The authors may also want to consider discussing if their methods/protocols have been previously used in similar studies or if they have been modified for this study.

Conclusion

The same it’s true for the conclusions paragraph. While again the conclusions are in agreement with the technical nature of this manuscript, I believe it would be helpful to have broader conclusions that discuss the significance of this work (even in a broader sense), that talks about potential future directions and talk about potential limitations (if any) to their protocols. 

Well done. Great work! 

Reviewer 3 Report

Comments and Suggestions for Authors

This study aims to evaluate and characterize the avipoxviruses circulating in Portugal between 2017 and 2023. The manuscript is written fluently. The figures are informative and clear. I see no limitations to the study.  

Reviewer 4 Report

Comments and Suggestions for Authors

Comments to authors

Overall, this manuscript is mostly well written and of relevance to the veterinary and scientific community. It provides important information on the avian pox viruses present in Portugal in recent years. However, the manuscript is very short and only adds 10 new DNA sequences. If possible, it would be beneficial to sequence additional samples to obtain a clearer picture of the diversity of avian pox viruses circulating in Portugal both recently and at earlier times in the past.

Major comments:

-          It is difficult to infer, with the data at hand, that there have been multiple introductions of avian pox viruses into Portugal and I would suggest presenting alternative hypotheses. This is especially the case due to the low sample size and limited time range from which the samples were collected. It is just as likely that the avian pox viruses observed in this study were already circulating in bird populations in Portugal for many years.

-        I would like to see some more discussion of the different subclades and what is known about them, and their importance in terms of virulence, etc (if that information is available). Please note that this is not my field of study so I am not aware of the literature that already exists.

Minor Comments:

                -Please make sure to cite all software used – Clustal software appears not to be cited.

Comments on the Quality of English Language

There are a few typos/grammatical errors – please revise carefully before publication.

Round 2

Reviewer 1 Report

Comments and Suggestions for Authors

Most of the comments raised in a previous round of the review has been addressed and the efforts from the Authors to revise this work is appreciated.

I have two items to be corrected (details below). After these are addressed, I would recommend the manuscript for publication.

First: Please check -still both versions of program MEGA X and MEGA 11 are in the manuscript text. As the tree is created the caption is created in parallel. Now in caption of Figure 3 MEGA 11 is placed – so it it appears that the version 11 is correct.

Second: Quality of figures 2 and  3 – resolution is not proper.

Author Response

Dear editor,

Thank you for your advices. Our responses are indicated below:

- First: Please check -still both versions of program MEGA X and MEGA 11 are in the manuscript text. As the tree is created the caption is created in parallel. Now in caption of Figure 3 MEGA 11 is placed – so it it appears that the version 11 is correct.

We apologize for did not correct properly this statement in the revised form of the manuscript. Indeed, we started to use MEGA X and then changed to MEGA 11, which originated the mistake.

Second: Quality of figures 2 and  3 – resolution is not proper.

Thank you for your advice. We tried to improve the resolution of the figures, but we are not sure we have achieved. Withou zoom, figures don't seem to be with high resolution, but when we zoom them in, we think that they are much better. The powerpoint files were also submitted apart from the manuscript. We hope that this will be enough.